# Treatment Patterns and Outcomes for Patients with Ampullary Carcinoma Who Do Not Undergo Surgery

**DOI:** 10.3390/cancers15143727

**Published:** 2023-07-22

**Authors:** Benjin D. Facer, Jordan M. Cloyd, Ashish Manne, Kenneth L. Pitter, Dayssy A. Diaz, Jose G. Bazan, Eric D. Miller

**Affiliations:** 1Department of Radiation Oncology, The Arthur G. James Cancer Hospital and Richard J. Solove Research Institute, The Ohio State University Comprehensive Cancer Center, Columbus, OH 43210, USA; 2Division of Surgical Oncology, The Arthur G. James Cancer Hospital and Richard J. Solove Research Institute, The Ohio State University Comprehensive Cancer Center, Columbus, OH 43210, USA; 3Department of Internal Medicine, Division of Medical Oncology, The Arthur G. James Cancer Hospital and Richard J. Solove Research Institute, The Ohio State University Comprehensive Cancer Center, Columbus, OH 43210, USA; 4Department of Radiation Oncology, City of Hope Comprehensive Cancer Center, Duarte, CA 91010, USA

**Keywords:** ampullary carcinoma, chemoradiation, gastrointestinal malignancy

## Abstract

**Simple Summary:**

Ampullary adenocarcinoma is a rare tumor in the gastrointestinal tract. Surgery is the preferred treatment, however if a patient has other medical conditions or advanced disease, surgery may not be possible. In this situation, the best treatment strategy is unknown. We sought to find out what happens to these patients in terms of treatments and survival. We used the National Cancer Database 2004–2017 to find 2176 patients who were diagnosed with ampullary adenocarcinoma but did not undergo surgery. The majority of these patients did not receive any chemotherapy or radiation. The rest received a combination of chemotherapy, palliative radiation, and/or definitive radiation. One-year overall survival ranged from 35% in patients who only received palliative radiation to 59.4% in patients who received chemotherapy and definitive radiation therapy. We did not find a significant difference in survival between patients who received chemotherapy and those who received chemotherapy and definitive radiation therapy.

**Abstract:**

Surgical resection is the standard of care for ampullary adenocarcinoma (AC). Many patients are ineligible due to comorbidities/advanced disease. Evidence for the optimal non-operative management of localized AC is lacking. We hypothesize that patients treated with chemotherapy (CT) and definitive radiation (DRT) will have superior survival (OS) compared to those treated with CT alone. We performed a retrospective review of the National Cancer Database from 2004 to 2017 to identify patients with non-metastatic AC and no surgical intervention. Patients were categorized as having received no treatment, palliative radiotherapy (PRT) alone, CT alone, CT + PRT, DRT alone, or CT + DRT. We utilized Kaplan–Meier analysis to determine OS and the log-rank test to compare survival curves. Among 2176 patients, treatment groups were: No treatment (71.2%), PRT alone (1.9%), CT alone (13.1%), CT + PRT (1.6%), DRT alone (2.4%), and CT + DRT (9.7%). One-year OS varied by treatment group, ranging from 35.1% (PRT alone) to 59.4% (CT + DRT). The one-year OS in a matched cohort was not significantly different between CT alone and CT + DRT (HR 0.87, 95% CI 0.69–1.10, *p* = 0.87). Most patients with non-metastatic AC not treated with surgery do not receive any treatment. There is no difference in one-year OS between those undergoing CT alone and CT + DRT.

## 1. Introduction

Ampullary adenocarcinoma (AC) is a rare cancer type, comprising less than 1% of gastrointestinal malignancies [1]. Surgical resection is the standard of care curative approach for localized cancers, but approximately 40–50% of patients are ineligible due to medical comorbidities or locally advanced disease [2,3,4]. Patients with localized AC who are able to undergo surgery have a one-year OS of greater than 90% [5,6]. The outcomes of patients with unresectable, non-metastatic disease are less clear. Although management typically consists of chemotherapy and/or radiotherapy according to institutional best-practices, no guidelines exist as to the optimal regimen, sequencing of therapies, or radiotherapy dosing. The utility of neoadjuvant and adjuvant therapies has been reported [3,5,6,7,8,9,10] but evidence for the optimal non-operative treatment of localized AC is lacking. Due to the rare nature of AC, prospective clinical trials are challenging to perform, underscoring the need for retrospective data with large patient numbers to identify treatment patterns and generate hypotheses for future studies. The primary objective of this study was to characterize practice patterns and survival rates for patients with AC unsuitable for surgical resection. A secondary objective was to compare the outcomes of non-operative strategies to potentially identify the optimal treatment strategy for this subset of patients. We hypothesized that patients treated with a combination of chemotherapy (CT) and definitive radiation therapy (DRT) would have superior one-year OS compared to those treated with CT alone.

## 2. Materials and Methods

### 2.1. Patient Selection

A retrospective review of the National Cancer Database (NCDB) from 2004 to 2017 was performed to identify patients ≥ 18 years old with non-metastatic ampullary adenocarcinoma. The NCDB is a combined effort of the Commission on Cancer (CoC) of the American College of Surgeons and the American Cancer Society and represents >70% of cancer diagnoses in the United States from over 1500 healthcare facilities [11]. The CoC’s NCDB and the hospitals participating in the CoC NCDB are the source of the deidentified data, and they have not verified and are not responsible for the statistical validity of the data analysis or the conclusions presented in this study. Use of the NCDB is not considered human subject research and approval by our institutional review board was deemed not necessary. Clinical, T, N, and M staging are defined using the criteria from the American Joint Committee on Cancer (AJCC) [12]. Histologic subtypes are classified according to World Health Organization criteria. A summary of exclusion criteria is provided in Figure 1.

In brief, all primary AC patients with metastatic disease, unknown clinical stage, non-adenocarcinoma histology, lack of follow-up, non-abdominal RT, unknown receipt of CT or RT, unknown RT dose, and non-definitive surgical intervention (e.g., debulking only) were excluded. Patients who underwent primary surgical intervention were included for portions of the analysis. Demographic information, including age, sex, year of diagnosis, distance to care, and race were obtained. An age ≥80 was defined as “older”. Socioeconomic status was estimated by matching the 2012 American Community Survey data against the patient’s home zip code [13] and stratified by less than or greater than/equal to the median income ($48,000). Comorbidity information was evaluated with Charlson/Deyo scoring and was recorded as no comorbidities or 1+ comorbidities. Treatment data included receipt of CT and RT, RT dose, number of RT fractions, and RT method (3D, IMRT, etc.).

### 2.2. Treatment Groups

We defined six treatment groups: no treatment, palliative radiotherapy (PRT) alone, CT alone, CT with PRT, DRT alone, and CT with DRT. PRT was defined as any course less than 45 Gy that was not otherwise classified as stereotactic body radiotherapy (SBRT). All RT regimens categorized as SBRT or those with a prescription of 45 Gy or more were classified as DRT. There was no time requirement between CT and RT for a patient to be included in the CT + PRT/DRT groups, although this association was explored in later analyses.

### 2.3. Statistical Methods

Logistic regression was utilized to analyze factors that may have been predictive of receiving different forms of treatment. All variables were extracted from NCDB data. Variables included age, gender (M vs. F), comorbidities (Charlson-Deyo 0 vs. 1+), race (white vs. non-white), income (<$48,000 annual income vs. ≥$48,000 annual income), year diagnosed (2004–2010 vs. 2011–2017), clinical stage (I vs. II/III), T-stage (1–2 vs. 3–4), N-stage (N0 vs. N1), and distance from site of treatment (<10.2 mi vs. ≥10.2 mi, 10.2 mi being the median). Logistic regression was performed using all variables to determine the odds of receiving treatment vs. no treatment, palliative treatment (PRT alone, CT alone, CT + PRT) vs. definitive treatment (DRT alone, CT + DRT), and CT vs. CT + DRT. One-year OS differences between groups were evaluated using Kaplan–Meier curves and compared using the log-rank test. Cox proportional hazards models were used to estimate adjusted hazard ratios (HRs) and 95% confidence intervals (CIs) for 1-year OS. Univariate analysis included age, sex, race, income, comorbidities, distance to care, year of diagnosis, and stage, as previously described. Variables with *p* < 0.10 on univariate analysis were included in the multivariable Cox model.

We also used a propensity score matched analysis (PSMA) to further control for confounding variables. This analysis included age, year of treatment, clinical stage, sex, race, comorbidities, income, and distance from care. Patients receiving CT alone were matched with those receiving CT + DRT using a 1:1 nearest available neighbor match without replacement using a caliper size calculated as 20% of the standard deviation of the propensity score [14]. Propensity score distributions were evaluated by computing the standardized difference of the covariates across the two groups. Following PSMA, OS between treatment groups was estimated using the Kaplan–Meier method and the effect of CRT was evaluated with a Cox proportional hazards model. All statistical tests were based on 2-sided probability with the significance level set at *p* < 0.05 using RStudio (RStudio, Inc., Boston, MA, USA) and SAS (SAS Institute Inc., Cary, NC, USA).

## 3. Results

### 3.1. Treatment Groups

A total of 10,797 patients with non-metastatic AC were included, of which 8621 patients (79.8%) underwent definitive surgical intervention and 2176 patients (20.1%) did not (Figure 1). The vast majority of patients who did not receive definitive surgery (n = 1550; 71.2%) did not receive any form of CT or RT (Figure 2A). Treatment groups included CT alone (n = 286; 13.1%), CT + DRT (n = 212; 9.7%), DRT alone (n = 52; 2.4%), PRT alone (n = 41; 1.9%), and CT + PRT (n = 35; 1.6%). Demographic and staging information are shown in Table 1.

The overall median age was 79 years. Men comprised 53.4% of the population. Almost all patients (85.2%) were non-Hispanic white. There was heterogeneity among treatment groups in terms of age, comorbidity status, years diagnosed, clinical stage, T stage, and N stage (Table 1). The incidence of CT alone increased and the incidence of RT use decreased with increasing stage (Figure 2B). For patients who received CT + DRT, 77.4% of patients started CT and DRT within 30 days of each other, 83.0% within 60 days, and 87.3% within 90 days, with 10.4% of patients starting CT and DRT greater than 90 days apart (2.4% with missing dates). A total of 340 patients received radiotherapy, 264 (77.6%) DRT, and 76 (22.3%) PRT. The most common DRT dose prescriptions were 45 Gy (n = 116, 34.1%) and 50.4 Gy (n = 69, 20.2%). Treatment technologies were categorized as IMRT (n = 107, 31.4%), 3D (n = 50, 14.7%), SBRT (n = 8, 2.4%), Other (n = 4, 1.2%), or Unknown (n = 171, 50.3%).

### 3.2. No Treatment vs. Treatment

We identified factors associated with receipt of treatment (Table 2). On multivariate analysis, patients were more likely to receive some type of treatment if they were diagnosed from 2011 to 2017, had clinical stage II or III disease, or had positive lymph nodes. They were less likely to receive treatment if they were 80 years or older or were female.

### 3.3. Palliative Therapy vs. Definitive Therapy

Factors associated with the receipt of receiving palliative therapy vs. definitive therapy were examined (Table 3). On multivariate analysis, older patients who received treatment were more likely than younger patients to receive definitive therapy. Younger patients were more likely to receive CT alone (classified as a palliative treatment) than older patients (Figure 3).

### 3.4. Chemotherapy Alone vs. Chemotherapy and Definitive Radiation Therapy

We also investigated factors associated with receiving CT vs. CT plus DRT (Table 4). On multivariate analysis, patients 80 years or older were more likely to receive CT + DRT than CT alone, again likely due to the relatively small proportion of older patients who received CT alone (Figure 3).

### 3.5. Overall Survival

Median survival for the entire cohort was 9.5 months, but survival varied by treatment type. One-year OS: No treatment 36.7%, PRT alone 35.1%, CT alone 53.4%, CT with PRT 45.7%, DRT alone 56.7%, CT with DRT 59.4% (Table 5, Figure 4A). Median OS: No treatment 7.9 mo, PRT alone 9.5 mo, CT alone 13.1 mo, CT with PRT 10.4 mo, DRT alone 14.7 mo, CT with DRT 13.7 mo. For reference, patients treated with definitive surgical intervention had a median OS of 49.5 months and a 1-year OS of 84.3% (Figure 4B). Patients who received any treatment had improved 1-year overall survival compared to those who did not receive any treatment (54.1% vs. 36.7%, *p* < 0.001). Patients treated with CT + DRT did not have a statistically significant improvement in 1-year OS compared to those who received CT alone (59.4% vs. 53.4%, *p* = 0.16). In addition, there was no statistically significant difference in 1-year OS among patients who received CT alone, DRT alone, or CT + DRT (53.4%, 56.7% and 59.4%, *p* = 0.18).

We evaluated factors associated with OS in those treated with CT alone vs. CT + DRT. On Cox univariate regression, advanced stage and farther distance to care were associated with worse 1-year OS (Table 6). Diagnosis after 2011 and unknown distance to care were associated with improved 1-year OS. Statistically significant variables, as well as age, were used in multivariable cox regression, which demonstrated no statistically significant difference in 1-year OS between patients with CT alone and CT + DRT (HR 0.86, 95% CI 0.72–1.08, *p* = 0.227). There was no significant association with stage, age, or unknown distance to care and OS. Patients who were diagnosed after 2011 had a decreased risk of death (HR 0.81, 95% CI 0.66–0.99, *p* = 0.039) and those who lived ≥ 10.2 miles from care had an increased risk of death (HR 1.23, 1.01–1.50, *p* = 0.043).

On PSMA, there were 185 matched pairs with a caliper width of 0.027 and standard deviation of 0.133. All baseline covariates were well-balanced between the two groups based on a standardized difference of ≤0.10. The median follow-up for the 370 patients in the matched cohort was 12.9 months. In this matched patient cohort, there was no statistically significant difference in 1-year OS between patients who received CT alone and CT + DRT (HR 0.87, 95% CI 0.69–1.10, *p* = 0.87) (Figure 5).

## 4. Discussion

This study found that the majority of patients with ampullary carcinoma diagnosed from 2004 to 2017 who did not undergo surgical resection did not receive CT or RT to the primary tumor. The underlying reasons for this trend are not ascertainable in a large, generalized dataset that does not capture all treatment decisions. Uncaptured factors include surgical fitness and willingness to undergo surgery, chemotherapy, or radiation. However, several influential covariates were available and included in the analysis. Of these covariates, year of diagnosis, clinical stage, lymph node status, sex, and age were all predictive of whether or not a patient would receive some form of CT or RT. The finding that most patients receive no treatment is important in light of the additional finding that patients who received some form of treatment were associated with improved survival compared to those who received none. This benefit is potentially tempered due to a selection bias in this population, in which only the most fit patients would be candidates for CT or RT. However, comorbidity status was included in this analysis and was not found to be predictive of survival.

This study also emphasizes the poor prognosis of patients who are unable to undergo surgical resection and adds to the sparse body of knowledge regarding non-operative management of AC. The ABC trial examined the role of cisplatin-gemcitabine vs. gemcitabine in advanced biliary tract carcinoma and included 20 patients with AC [15]. A cohort of AC tumors combined with bile duct tumors demonstrated a poor overall response rate to chemotherapy (<20%). No significant difference was found between chemotherapy groups for AC (HR 0.62, 95% CI 0.21–1.82). Rostain et al. studied patients with AC diagnosed from 1976 to 2009; specific non-operative treatment strategies were not described, but 1-year OS was reported to be 26.5% [3].

Based on our analysis, approximately 20% of patients with non-metastatic AC underwent primary non-operative therapy. As a combined cohort, these patients had worse 1-year OS compared to the surgical patients (41.8% vs. 84.3%, *p* < 0.001) (Figure 4B). These numbers are comparable to a population-based analysis from the Netherlands, which demonstrated 1-year OS for non-metastatic AC as approximately 80% with resection and 40% without resection [4]. In that cohort, nonsurgical patients receiving chemotherapy and/or radiotherapy comprised <1% of all non-metastatic patients, compared to 5.8% of our cohort.

In terms of survival, patients treated with CT + DRT did not have improved one-year OS compared to those with CT alone. This observation remained consistent after PSMA. The LAP07 trial demonstrated a similar result [16], in which patients with unresectable pancreatic cancer, an anatomically similar but different prognosis malignancy, were randomized to receive concurrent chemoradiation or chemotherapy alone. There was no difference in median OS between groups, though there was decreased local progression in the chemoradiation group. However, in the present study, a third group was also analyzed, DRT alone. When DRT was compared to CT alone and CT + DRT, there was no difference in 1-year OS. This may suggest that patients with unresectable AC may gain a similar benefit from either CT or DRT alone. Similar outcomes across multiple treatment strategies support the notion that treatment decisions for rare malignancies should be made in a multidisciplinary setting so that management can be personalized for each patient. For example, a patient who is not medically fit for surgery or chemotherapy but is willing to come for daily RT should be given that option with a full understanding that her chances of survival are not worse, based on our findings, because of the inability to receive chemotherapy. In another situation, a patient living far from medical care may not want to pursue radiotherapy, as it could be a large financial and time burden, so chemotherapy alone may be a better option. The relatively similar OS among all three treatment modalities offers multiple options for the clinician to personalize care for the patient without compromising outcomes.

This study has several limitations. As previously mentioned, the NCDB does not include sufficient detail to ascertain why patients receive the prescribed treatment modalities. This was mitigated in part through the utilization of multivariate regression. In addition, we performed PSMA for the two largest groups of patients who received treatment. This analysis confirmed a lack of statistical difference in OS between these two groups. Secondly, the NCDB does not have cause-specific survival data, so our analysis of survival is limited to OS. However, in the setting of an aggressive cancer with a relatively poor prognosis, the difference between CSS and OS is not as distinct as for malignancies with a better prognosis. Third, retrospective data is fraught with various forms of bias that cannot be corrected without a prospective, randomized trial. Due to the rare nature of AC, prospective trials are not likely to accrue well, and we must draw conclusions based on the data that is available. Fourth, although not a primary focus of this study, the extent of surgery (Whipple, ampullectomy, etc.) is not available in the NCDB, which somewhat limits the direct comparison of these data to any specific surgical procedure. Quality of life measures are other important factors not captured in the NCDB and are missing from this study. Lastly, there is significant heterogeneity among these treatments that is not adequately captured. For example, the type, dosing and frequency of CT is not reported in the NCDB, but survival may vary between approaches. As general treatment paradigms become more established, single- or multi-institutional studies will indicate optimal CT and RT approaches.

## 5. Conclusions

This is the first NCDB study demonstrating treatment and overall survival patterns for the non-surgical management of localized AC. The majority of patients received no cancer-directed therapy, which was associated with a poor prognosis. Those who were able to receive some form of treatment were associated with improved OS. We found no significant difference in 1-year OS among patients who received CT alone, DRT alone, or CT + DRT. Patients with AC who cannot undergo surgical resection should be evaluated by a multidisciplinary team to form a personalized treatment plan.

## Figures and Tables

**Figure 1 cancers-15-03727-f001:**
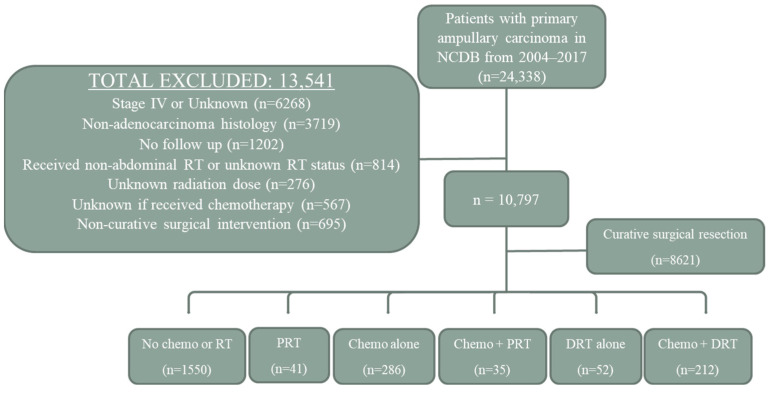
Flow chart of patient selection. Abbreviations: NCDB, National Cancer Database; RT, radiotherapy; Chemo, chemotherapy; PRT, palliative radiotherapy; DRT, definitive radiotherapy.

**Figure 2 cancers-15-03727-f002:**
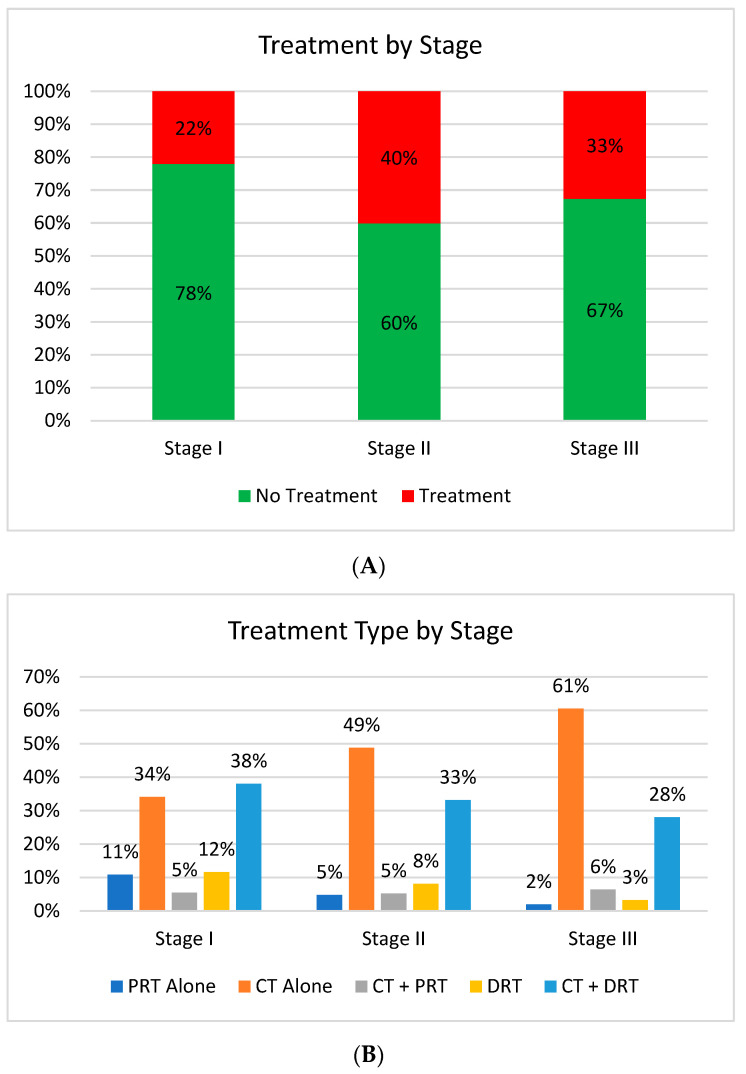
(**A**) Treatment by stage (including patients with no treatment). (**B**) Treatment by stage (not including patients with no treatment). Abbreviations: CT, chemotherapy; PRT, palliative radiotherapy; DRT, definitive radiotherapy.

**Figure 3 cancers-15-03727-f003:**
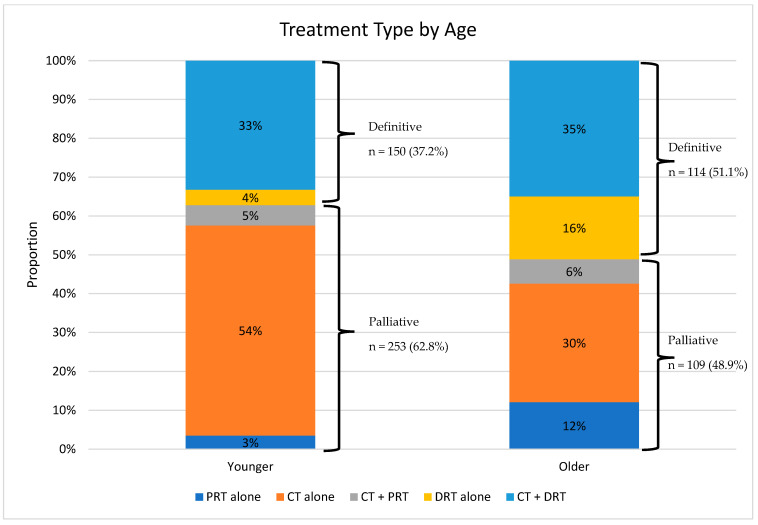
Treatment type by age (Younger = 79 and younger, Older = 80 and older). Abbreviations: CT, chemotherapy; PRT, palliative radiotherapy; DRT, definitive radiotherapy.

**Figure 4 cancers-15-03727-f004:**
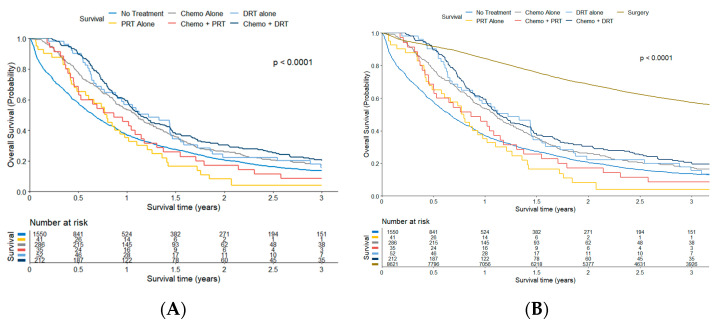
(**A**): Overall survival by treatment group. (**B**): Overall survival with surgical patients included. Abbreviations: Chemo, chemotherapy; PRT, palliative radiotherapy; DRT, definitive radiotherapy.

**Figure 5 cancers-15-03727-f005:**
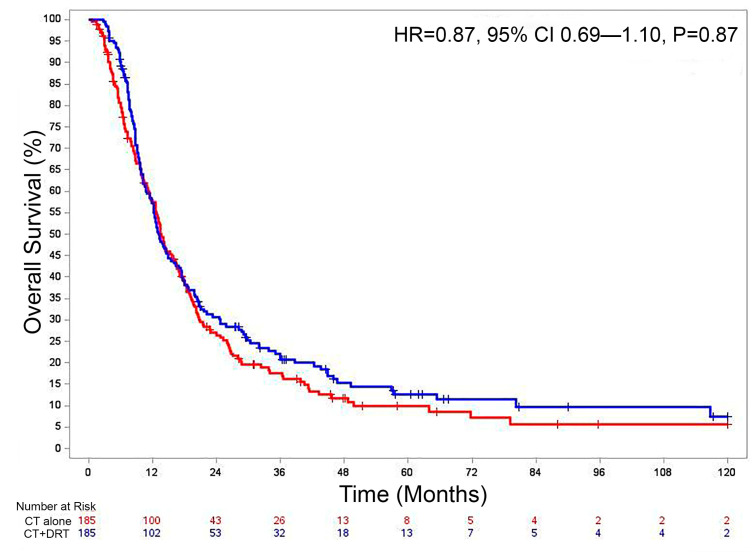
Overall survival for propensity-matched cohort for CT alone vs. CT + DRT. Abbreviations: CT, chemotherapy; DRT, definitive radiotherapy.

**Table 1 cancers-15-03727-t001:** Patient characteristics by treatment group.

	No Treatment		PRT Alone		CT Alone		CT + PRT		DRT Alone		CT + DRT	
	N	%	N	%	N	%	N	%	N	%	N	%
Patients	1550	71.2%	41	1.9%	286	13.1%	35	1.6%	52	2.4%	212	9.7%
Mean age	77.8		80.2		69.6		74.9		81.7		73.3	
Age groups												
79 or younger	694	44.8%	14	34.1%	218	76.2%	21	60.0%	16	30.8%	134	63.2%
80 or older	856	55.2%	27	65.9%	68	23.8%	14	40.0%	36	69.2%	78	36.8%
Mean follow-up (months)	14.8		10.9		18.5		13.7		20.0		21.8	
Gender												
Male	780	50.3%	27	65.9%	172	60.1%	22	62.9%	31	59.6%	129	60.8%
Female	770	49.7%	14	34.1%	114	39.9%	13	37.1%	21	40.4%	83	39.2%
Charlson-Deyo Comorbidity Score												
0	935	60.3%	22	53.7%	191	66.8%	16	45.7%	28	53.8%	142	67.0%
1+	615	39.7%	19	46.3%	95	33.2%	19	54.3%	24	46.2%	70	33.0%
Distance from care												
<10.2 mi	738	47.6%	25	61.0%	130	45.5%	21	60.0%	27	51.9%	90	42.5%
≥10.2 mi	747	48.2%	12	29.3%	142	49.7%	14	40.0%	24	46.2%	107	50.5%
Unknown	65	4.2%	4	9.8%	14	4.9%	0	0.0%	1	1.9%	15	7.1%
Race												
White	1321	85.2%	37	90.2%	238	83.2%	29	82.9%	44	84.6%	185	87.3%
Non-White	229	14.8%	4	9.8%	48	16.8%	6	17.1%	8	15.4%	27	12.7%
Income												
<$48,000 per year	653	42.1%	16	39.0%	115	40.2%	19	54.3%	19	36.5%	87	41.0%
≥$48,000 per year	826	53.3%	21	51.2%	157	54.9%	16	45.7%	32	61.5%	109	51.4%
Unknown	71	4.6%	4	9.8%	14	4.9%	0	0.0%	1	1.9%	16	7.5%
Year diagnosed												
2004–2010	680	43.9%	10	24.4%	81	28.3%	12	34.3%	29	55.8%	91	42.9%
2011–2017	870	56.1%	31	75.6%	205	71.7%	23	65.7%	23	44.2%	121	57.1%
Clinical Stage												
I	911	58.8%	28	68.3%	88	30.8%	14	40.0%	30	57.7%	98	46.2%
II or III	639	41.2%	13	31.7%	198	69.2%	21	60.0%	22	42.3%	114	53.8%
T Stage												
1 or 2	646	41.7%	24	58.5%	105	36.7%	12	34.3%	17	32.7%	76	35.8%
3 or 4	385	24.8%	10	24.4%	125	43.7%	15	42.9%	10	19.2%	58	27.4%
Unknown	519	33.5%	7	17.1%	56	19.6%	8	22.9%	25	48.1%	78	36.8%
N Stage												
0	892	57.5%	29	70.7%	152	53.1%	19	54.3%	24	46.2%	101	47.6%
1	119	7.7%	5	12.2%	74	25.9%	8	22.9%	3	5.8%	33	15.6%
Unknown	539	34.8%	7	17.1%	60	21.0%	8	22.9%	25	48.1%	78	36.8%

CT = chemotherapy, PRT = palliative radiotherapy, DRT = definitive radiotherapy.

**Table 2 cancers-15-03727-t002:** Logistic regression of factors associated with no treatment vs. treatment.

	Univariate				Multivariate			
Variable	OR	Lower CI	Upper CI	*p*-Value	OR	Lower CI	Upper CI	*p*-Value
Age: ≥80 vs. <80	0.85	0.82	0.88	0.000	0.88	0.84	0.92	0.000
Sex: F vs. M	0.92	0.88	0.95	0.000	0.94	0.90	0.99	0.012
Race: non-white vs. white	1.00	0.95	1.06	0.961				
Income: ≥$48,000 vs. <$48,000	1.01	0.97	1.05	0.730				
Comorbidity: ≥1 vs. 0	0.97	0.93	1.01	0.140				
Distance to care: <10.2 mi vs. ≥10.2 mi	1.00	1.00	0.96	1.040				
Year of diagnosis: 2011–2017 vs. 2004–2010	1.07	1.03	1.11	0.000	1.07	1.01	1.01	0.026
Stage II–III vs. Stage I	1.16	1.11	1.20	0.000	1.13	1.00	1.27	0.049
T3–4 vs. T1–2	1.10	1.05	1.15	0.000	0.96	0.86	1.07	0.442
N1 vs. N0	1.27	1.20	1.36	0.000	1.16	1.07	1.26	0.000

**Table 3 cancers-15-03727-t003:** Logistic regression analysis of factors associated with palliative vs. definitive treatment.

	Univariate				Multivariate			
Variable	OR	Lower CI	Upper CI	*p*-Value	OR	Lower CI	Upper CI	*p*-Value
Age: ≥80 vs. <80	1.15	1.06	1.25	0.001	1.19	1.09	1.31	0.000
Sex: F vs. M	1.00	0.93	1.09	0.911				
Race: non-white vs. white	0.95	0.85	1.06	0.338				
Income: ≥$48,000 vs. <$48,000	1.01	0.94	1.10	0.747				
Comorbidity: ≥1 vs. 0	0.99	0.91	1.07	0.771				
Distance to care: <10.2 mi vs. ≥10.2 mi	1.04	0.96	1.13	0.339				
Year of diagnosis: 2011–2017 vs. 2004–2010	0.83	0.77	0.90	0.000	1.01	0.88	1.16	0.916
Stage II–III vs. Stage I	0.88	0.81	0.95	0.002	0.94	0.78	1.14	0.531
T3–4 vs. T1–2	0.92	0.84	1.00	0.058	0.98	0.83	1.15	0.815
N1 vs. N0	0.91	0.83	1.01	0.071	0.98	0.85	1.12	0.746

**Table 4 cancers-15-03727-t004:** Logistic Regression of Factors Associated with CT Alone vs CT + DRT.

	Univariate				Multivariate			
Variable	OR	Lower CI	Upper CI	p-Value	OR	Lower CI	Upper CI	*p*-Value
Age: ≥80 vs. <80	1.16	1.06	1.28	0.002	1.24	1.11	1.38	0.000
Sex: F vs. M	0.99	0.91	1.09	0.873				
Race: non-white vs. white	0.93	0.82	1.05	0.212				
Income: ≥$48,000 vs. <$48,000	0.99	0.91	1.08	0.853				
Comorbidity: ≥1 vs. 0	1.00	0.91	1.09	0.963				
Distance to care: <10.2 mi vs. ≥10.2 mi	1.02	0.93	1.12	0.652				
Year of diagnosis: 2011–2017 vs. 2004–2010	0.85	0.78	0.93	0.001	1.04	0.90	1.21	0.583
Stage II–III vs. Stage I	0.85	0.78	0.93	0.040	0.87	0.75	1.02	0.093
T3–4 vs. T1–2	0.90	0.82	1.00	0.042	1.01	0.87	1.17	0.895
N1 vs. N0	0.91	0.82	1.02	0.104				

**Table 5 cancers-15-03727-t005:** Overall survival by treatment group.

Treatment Groups	Median OS (mo)	1 yr OS	3 yr OS
No Treatment	7.9	36.7%	13.6%
PRT alone	9.5	35.1%	4.1%
CT alone	13.1	53.4%	17.3%
CT + PRT	10.4	45.7%	8.6%
DRT alone	14.7	56.7%	15.6%
CT + DRT	13.7	59.4%	20.1%

CT = chemotherapy, PRT = palliative radiotherapy, DRT = definitive radiotherapy.

**Table 6 cancers-15-03727-t006:** Univariate and multivariate Cox regression analysis for overall survival for CT + DRT vs. CT alone.

	Univariate Analysis				Multivariate Analysis			
Variable	HR	Lower CI	Upper CI	*p*-Value	HR	Lower CI	Upper CI	*p*-Value
CT + DRT vs. CT Alone	0.86	0.71	1.05	0.139	0.88	0.72	1.08	0.227
Year of Diagnosis: 2011–2017 vs. 2004–2010	0.79	0.65	0.97	0.021	0.81	0.66	0.99	0.039
Stage II–III vs. Stage I	1.24	1.01	1.51	0.038	1.22	0.99	1.50	0.063
Age (continuous)	1.00	0.99	1.01	0.992	1.01	1.00	1.01	0.262
Sex: F vs. M	1.07	0.88	1.30	0.503				
Race: Non-white vs. white	0.81	0.62	1.06	0.129				
Comorbidity: ≥1 vs. 0	1.06	0.86	1.29	0.599				
Income: ≥$48,000 vs. <$48,000	0.87	0.72	1.05	0.153				
Distance to care								
<10.2 mi	Ref	Ref	Ref	Ref				
≥10.2 mi	1.23	1.01	1.50	0.041	1.23	1.01	1.50	0.043
Unknown	0.60	0.38	0.97	0.036	0.67	0.41	1.08	0.097

CT = chemotherapy, DRT = definitive radiotherapy.

## Data Availability

All data are from the National Cancer Database records.

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
