# Peer review of "Treatment Patterns and Outcomes for Patients with Ampullary Carcinoma Who Do Not Undergo Surgery"

_cancers, 2023, doi:10.3390/cancers15143727_

Round 1

Reviewer 1 Report

This is a good use of the NCDB  for a question that will not be likely to be answered in a prospective study.

It is retrospective and tells us what is currently (2004 -2017) done and what is the survival of these patients.  And here comes the main problem: Patients that are not eligeable  for surgery usually have important co-morbidities.  Would it be possible to read more about cancer related mortality and no cancer related mortality?

How do the autors interprete the survival curve of the no treatment group that crosses the yellow and the orange line after about one to 1.5 years?

Minor: In figure 2a: use other colours, that do not re-appear in the further graphs. 

Author Response

Reviewer #1:

This is a good use of the NCDB for a question that will not be likely to be answered in a prospective study.

Comment 1:

It is retrospective and tells us what is currently (2004 -2017) done and what is the survival of these patients.  And here comes the main problem: Patients that are not eligeable  for surgery usually have important co-morbidities.  Would it be possible to read more about cancer related mortality and no cancer related mortality?

Our response: Thank you for this insightful comment.  We agree that patients not eligible for surgical resection likely have some associated medical comorbidity which excludes them from standard-of-care treatment.  Unfortunately, the NCDB does not provide cause-specific survival data, but only overall survival.  Thus, we agree that an analysis of cancer-related mortality would add significantly to the results of this analysis, but is not possible given the limitations of the NCDB.  We have included this limitation in the last paragraph of the discussion: “Secondly, the NCDB does not have cause-specific survival data, so our analysis of survival is limited to OS.”

Comment 2:

How do the autors interprete the survival curve of the no treatment group that crosses the yellow and the orange line after about one to 1.5 years?

Our response: Thank you for this observation.  The reviewer does make an important point about the survival curve for all treatment groups including no treatment shown in Figure 4.  At 1 year, the no treatment group crosses the palliative radiation therapy (PRT) alone curve and then crosses the chemotherapy plus palliative radiation (Chemo+PRT) therapy alone curve at 1.5 years.  It is reassuring that the no treatment group survival curve lies below all of the other curves for the first year.  At 1 year, the number of patients in the PRT and Chemo+PRT is quite limited, so any interpretation of the results past that point must be taken with caution.  In addition, the no-treatment group started with 1550 patients while the PRT Alone and Chemo+PRT groups contained 41 and 35 patients, respectively.  Due to the above limitations, it is an interesting finding, but must be taken with caution given the limited number of patients that are being compared in those cohorts.

Comment 3:

Minor: In figure 2a: use other colours, that do not re-appear in the further graphs. 

This has been corrected.

Reviewer 2 Report

The study is well-done. I miss some technical details from the patient group surgically treated, like the type of the procedures (Whipple, ampullectomy, palliative measures?).

Author Response

Comment 1:

The study is well-done. I miss some technical details from the patient group surgically treated, like the type of the procedures (Whipple, ampullectomy, palliative ?).

Our response: Thank you for this insightful comment.  We agree that additional details regarding the surgical procedures utilized for those patients who were eligible would be helpful.  Unfortunately, details regarding surgery type are limited in the NCDB, thus, we were unable to provide additional information about the procedures used for the patients included in this study.  This is a shortcoming of utilizing the NCDB. The following line has been added in the last paragraph of the discussion: “Fourth, although not a primary focus of this study, the extent of surgery (Whipple, ampullectomy, etc.) is not available in the NCDB, which somewhat limits the direct comparison of these data to any specific surgical procedure.”

Reviewer 3 Report

It is an interesting paper trying to access the outcome of patients suffering an ampullary carcinoma but not having received surgery. Authors, based on the National Cancer Database [2004-2017] try to identify the "best" non-surgical manipulation [no treatment, palliative radiotherapy alone, chemotherapy alone, chemotherapy with palliative radiotherapy, definitive radiotherapy alone and chemotherapy with definitive radiotherapy] of the group of 2176 such patients.

From the practical poin of view, authors' findings serve as alternative options of non-surgical treatment, see lines 266-267 "a patient who is not medically fit for surgery or chemotherapy but is  willing to come for daily radiotherapy should be given that option ... a patient living far from medical care may not want to pursue radiotherapy, as it could be a large financial and time burden, so chemotherapy alone may be a better option"

On the other hand, the paper is based on data of year 2004-2017. Although authors split patients into two study periods the there are differences within  each treatment as the time pass.

Line 187: For reference, patients treated with definitive surgical  intervention had a median OS of 49.5 months and a 1-year OS of 84.3% (Figure S1). I prefer this figure to be within the text and not as supplement; allthough this information is not the purpose of the study, I need it for comparison. Thus, the significant difference  must be emphasized in discussion [lines 249-251]

I understand that data on QoL are not available in the National Cancer Dapabase. However, this is a limitation of the study and should be mentioned as such.

Figures 1-3 should be more professional: there is no contrast between grey background and while letters; the same for fig 2, black [small letters in blue]. Additionally, fig 2 and 3, titles within the figures are extremely large [20p?] in relation to the letters and numbers in the columns

Author Response

It is an interesting paper trying to access the outcome of patients suffering an ampullary carcinoma but not having received surgery. Authors, based on the National Cancer Database [2004-2017] try to identify the "best" non-surgical manipulation [no treatment, palliative radiotherapy alone, chemotherapy alone, chemotherapy with palliative radiotherapy, definitive radiotherapy alone and chemotherapy with definitive radiotherapy] of the group of 2176 such patients.

From the practical poin of view, authors' findings serve as alternative options of non-surgical treatment, see lines 266-267 "a patient who is not medically fit for surgery or chemotherapy but is  willing to come for daily radiotherapy should be given that option ... a patient living far from medical care may not want to pursue radiotherapy, as it could be a large financial and time burden, so chemotherapy alone may be a better option"

On the other hand, the paper is based on data of year 2004-2017. Although authors split patients into two study periods the there are differences within  each treatment as the time pass.

Comment 1:

Line 187: For reference, patients treated with definitive surgical  intervention had a median OS of 49.5 months and a 1-year OS of 84.3% (Figure S1). I prefer this figure to be within the text and not as supplement; allthough this information is not the purpose of the study, I need it for comparison. Thus, the significant difference  must be emphasized in discussion [lines 249-251]

Our response: Thank you for this suggestion. We have included Figure S1 as Figure 4B and added a reference to Figure 4B in the discussion

Comment 2:

I understand that data on QoL are not available in the National Cancer Dapabase. However, this is a limitation of the study and should be mentioned as such.

Our response: We agree, and this has been listed as a limitation in the discussion section: “Quality of life measures are other important factors not captured in the NCDB and are missing from this study.”

Comment 3:

Figures 1-3 should be more professional: there is no contrast between grey background and while letters; the same for fig 2, black [small letters in blue]. Additionally, fig 2 and 3, titles within the figures are extremely large [20p?] in relation to the letters and numbers in the columns

Our response: Thank you for these comments.  We have adjusted the appearances of Figures 1-3 to make them easier to read and more professional appearing.